

# Spatial structure arising from neighbour-dependent bias in collective cell movement

Rachelle N. Binny[1,2,6], Parvathi Haridas[3], Alex James[1,2], Richard Law[4], Matthew J. Simpson[3,5] and Michael J. Plank[1,2]

[1] School of Mathematics and Statistics, University of Canterbury, Christchurch, New Zealand
[2] Te Pūnaha Matatini, New Zealand
[3] Institute of Health and Biomedical Innovation, Queensland University of Technology, Brisbane, Australia
[4] York Centre for Complex Systems Analysis, Ron Cooke Hub, University of York, York, United Kingdom
[5] School of Mathematical Sciences, Queensland University of Technology, Brisbane, Australia
[6] Landcare Research—Manaaki Whenua, Lincoln, New Zealand

Corresponding author
Rachelle N. Binny,
binnyr@landcareresearch.co.nz

## ABSTRACT

Mathematical models of collective cell movement often neglect the effects of spatial structure, such as clustering, on the population dynamics. Typically, they assume that individuals interact with one another in proportion to their average density (the mean-field assumption) which means that cell–cell interactions occurring over short spatial ranges are not accounted for. However, *in vitro* cell culture studies have shown that spatial correlations can play an important role in determining collective behaviour. Here, we take a combined experimental and modelling approach to explore how individual-level interactions give rise to spatial structure in a moving cell population. Using imaging data from *in vitro* experiments, we quantify the extent of spatial structure in a population of 3T3 fibroblast cells. To understand how this spatial structure arises, we develop a lattice-free individual-based model (IBM) and simulate cell movement in two spatial dimensions. Our model allows an individual's direction of movement to be affected by interactions with other cells in its neighbourhood, providing insights into how directional bias generates spatial structure. We consider how this behaviour scales up to the population level by using the IBM to derive a continuum description in terms of the dynamics of spatial moments. In particular, we account for spatial correlations between cells by considering dynamics of the second spatial moment (the average density of pairs of cells). Our numerical results suggest that the moment dynamics description can provide a good approximation to averaged simulation results from the underlying IBM. Using our *in vitro* data, we estimate parameters for the model and show that it can generate similar spatial structure to that observed in a 3T3 fibroblast cell population.

## INTRODUCTION

Collective cell movement is integral to tissue repair (*Martin, 1997*; *Shaw & Martin, 2009*), embryonic development (*Kurosaka & Kashina, 2008*), the immune response (*Rørth, 2009*) and cancer (*Friedl & Wolf, 2003*). Interactions occurring between individual cells have implications for movement of the cell population as a whole. However, the manner in which these individual-level events affect the collective dynamics is not always well understood (*Tambe et al., 2011*; *Vedel et al., 2013*; *Agnew et al., 2014*). Cells interact over short length scales in various ways, for example via cell-secreted diffusible chemical signals (*Mason, Ito & Corfas, 2001*; *Raz & Mahabaleshwar, 2009*). When detected by neighbouring cells these signals can have a repulsive or attractive effect on an individual's direction of movement (*Painter & Hillen, 2002*), or affect the rate at which a cell will move (*Cai, Landman & Hughes, 2006*). Physical forces, such as cell–cell adhesion (*Trepat et al., 2009*; *Tambe et al., 2011*), and crowding effects also influence movement (*Abercrombie, 1979*; *Plank & Simpson, 2012*). These interactions may generate spatial structure in a cell population which will in turn affect the collective dynamics (*Plank & Law, 2015*). For instance, cell clustering can arise due to attractive forces such as cell–cell adhesion (*Green et al., 2010*; *Agnew et al., 2014*). On the other hand, repulsive forces such as chemorepellant signals can cause cells to segregate (*Kay, Chu & Sanes, 2012*; *Keeley et al., 2014*).

Individual-based models (IBMs) have proven effective for simulating the movement of large numbers of cells and can give insights into how interactions give rise to spatial structure (*Grimm et al., 2006*). In a lattice-free framework, cells are represented as individual agents undergoing movement through continuous space and features including proliferation (*Plank & Simpson, 2012*), cell–cell adhesion (*Johnston, Simpson & Plank, 2013*) and directional bias (*Dyson & Baker, 2015*) can be incorporated into the model. Equivalent lattice-based models, where agent locations are restricted to discrete sites on a pre-defined lattice, often require less computational power than their lattice-free counterparts. However, at high cell densities agents become aligned along the lattice resulting in unrealistic spatial configurations of cells that do not correspond well to those observed experimentally (*Plank & Simpson, 2012*). In lattice-free models, different approaches can be employed to account for crowding effects and volume-exclusion, the concept that the cells themselves take up space in the domain and may obstruct the movement of neighbouring cells. For instance, each individual may occupy a spherical region with fixed diameter through which the movement of other agents is restricted (*Bruna & Chapman, 2012*; *Dyson & Baker, 2015*).

IBMs for cell movement in two spatial dimensions generate simulation data that can be compared to experimental images of moving cells studied *in vitro*. In two-dimensional cell migration assays, such as circular barrier assays (*Simpson et al., 2013b*) and scratch assays (*Johnston, Simpson & McElwain, 2014*), cells are seeded into a well and allowed to attach to the well surface. The movement of cells across the surface can then be monitored by imaging the well at regular discrete time intervals. Analysis of this time-lapse imaging data provides information about the properties of individual cells as well as the spatial distribution of the population over time (*Simpson, Landman & Hughes, 2010*).

Using an IBM to obtain a reliable description of average cell behaviour can become computationally expensive because this involves carrying out many simulation repeats. In addition, IBMs are not particularly amenable to further mathematical analysis. This has motivated the development of more mathematically tractable approximation schemes which can provide greater insight into how population-level behaviour arises from interactions in the underlying stochastic process (*Deroulers et al., 2009*). Models that aim to capture collective movement at the population level, such as the Fisher–Kolmogorov equation (*Fisher, 1937*; *Kolmogorov, Petrovsky & Piskunov, 1937*), typically do not account for spatial structure. The majority of models invoke a mean-field assumption which assumes that cells interact with one another in proportion to their average density (*Anderson & Chaplain, 1998*; *Deroulers et al., 2009*; *Tremel et al., 2009*). Thus, they do not always provide an accurate representation of cell behaviour, particularly in highly clustered (or segregated) populations where interactions between neighbouring cells are often stronger (or weaker) than in populations where there is no spatial structure (*Simpson et al., 2013a*; *Markham, Baker & Maini, 2014*).

An alternative approach incorporates spatial correlations by employing the dynamics of spatial moments. The dynamics of individual cells, pair of cells, triplets of cells, and so on, can be considered in order to explore how spatial structure changes over time. In ecology, spatial moment models have been developed to study the effects of spatial patterns in animal and plant communities (*Bolker & Pacala, 1997*; *Lewis & Pacala, 2000*; *Dieckmann & Law, 2000*). Models incorporating birth, death (*Bolker & Pacala, 1997*; *Law, Murrell & Dieckmann, 2003*), growth (*Adams et al., 2013*) and movement (*Murrell & Law, 2000*) have been considered, as well as interactions between different types or species, for example predator–prey relationships (*Murrell, 2005*). More recently, moment dynamics approaches have also been applied to collective cell movement, such as in lattice-free models with chemotactic interactions (*Newman & Grima, 2004*; *Binny, Plank & James, 2015*) and cell–cell adhesion (*Middleton, Fleck & Grima, 2014*), and a lattice-based model for interacting cell populations (*Johnston, Simpson & Baker, 2015*).

A closure assumption is required in order to solve a dynamical system of spatial moments. The mean-field assumption closes the system at first order so ignores the spatial information held in higher moments. In order to retain information about spatial structure a second-order closure, at least, is needed. A number of different second-order closures are possible (*Murrell, Dieckmann & Law, 2004*; *Raghib, Hill & Dieckmann, 2011*); however, the Kirkwood Superposition Approximation is often applied in the context of cell movement (*Kirkwood, 1935*; *Kirkwood & Boggs, 1942*; *Markham, Baker & Maini, 2014*). Other schemes which do not rely on a closure assumption have also been developed, for example perturbation approximations (*Bruna & Chapman, 2012*) and methods that deal with spatial moments at all orders (*Ovaskainen et al., 2014*).

In this paper we extend the model described in our recent work (*Binny, Plank & James, 2015*) from one to two spatial dimensions, making it more amenable for use in conjunction with experimental data. To explore whether our model can provide insights into the behaviour of moving cells studied *in vitro*, we analyse imaging data generated from experiments with populations of motile 3T3 murine fibroblast cells.

We present a lattice-free IBM for collective cell movement in which an individual's rate and direction of movement are determined by interactions with cells in its neighbourhood. This neighbour-dependent directional bias allows us to explore how attractive or repulsive interactions between cells give rise to spatial structure in the population. The first spatial moment, the average density of individual cells, holds no spatial information. Therefore, in order to account for spatial correlations we consider the second spatial moment, an average density of pairs of cells. We use our IBM to derive a population-level description for the second moment dynamics and solve this for a distribution of cells that is homogeneous in space. Our results suggest that the spatial moment model can provide a good approximation to the underlying stochastic process.

Motile cells possess dynamic cytoskeletons which allow them to change their shape and flex around neighbouring cells (*Abercrombie, 1979*; *Le Clainche & Carlier, 2008*). To try and capture this trait we also make use of the neighbourhood-dependent directional bias as a mechanism for incorporating crowding effects, rather than defining cells as hard spheres with a fixed exclusion area. Using our *in vitro* data, we estimate parameters for the model and quantify the spatial structure in a moving population of fibroblast cells.

## EXPERIMENTAL METHODS

### Cell culture

Murine fibroblast 3T3 cells were cultured in Dulbecco's modified Eagle medium (Invitrogen, Australia) with 5% foetal calf serum (FCS) (Hyclone, New Zealand), 2 mM L-glutamine (Invitrogen, Carlsbad, CA, USA), 50 U/ml penicillin and 50 µg/ml streptomycin (Invitrogen), in 5% $CO_2$ and 95% air at 37 °C. Monolayers of 3T3 cells were cultured in T175 $cm^2$ tissue culture flasks (Nunc, Thermo Scientific, Denmark). Prior to confluence, cells were lifted with 0.05% trypsin (Invitrogen, Carlsbad, CA, USA). Viable cells were counted using the trypan blue exclusion test and a haemocytometer.

Two cell suspensions were created at approximate average cell densities of 20,000 cells/ml and 30,000 cells/ml. The experiments were performed in triplicate for each initial cell density. Cells were seeded in a 24 well tissue culture plate (each well of diameter 15.6 mm) and incubated overnight in 5% $CO_2$ and 95% air at 37 °C to allow them to attach to the base of the plate. Initially, cells were approximately uniformly distributed in each well.

### Imaging techniques and analysis

Time-lapse images of the cells were captured, over a period of 12 h at 3 h intervals, using a light microscope and Eclipse TIS software at 100× magnification. For each sample, a 4,500 µm × 450 µm image was reconstructed from overlapping adjacent images captured at approximately the centre of the well. The locations of the $n$ cells in each image were manually determined by superimposing markers onto cells and recording the Cartesian coordinates of markers using ImageJ image analysis software. These coordinates were used to calculate a pair-correlation function (PCF) for each image following the method in 'Pair-correlation function'.

# MATHEMATICAL MODELLING OF CELL MOVEMENT

## Individual-based model

We extend our previous model (*Binny, Plank & James, 2015*) to consider the collective movement of $n$ individuals in two-dimensional continuous space, with periodic conditions at the boundaries. The following framework is analogous to the one-dimensional model described in *Binny, Plank & James (2015)* and we refer the reader there for a more comprehensive description of the concepts outlined below.

The location of a cell $i$ is represented by a coordinate $\mathbf{x}_i \in \mathbb{R}^2$ and the state of the system at time $t$ comprises the locations of all $n$ individuals. Cell $i$ moves as a Poisson process over time with movement rate per unit time $\psi_i(\mathbf{x})$, i.e., the probability of an event occurring in a short time $\delta t$ is $\psi_i(\mathbf{x})\delta t + O(\delta t^2)$. The movement rate $\psi_i(\mathbf{x})$ is dependent on the state of the system at time $t$ so the Poisson process is inhomogeneous over time. When cell $i$ undergoes a movement event, it moves a displacement $\mathbf{r}$ to a new location $\mathbf{x}_i + \mathbf{r}$ drawn from a probability density function (PDF) $\mu(\mathbf{x}_i, \mathbf{x}_i + \mathbf{r})$.

We use the Gillespie algorithm to simulate this stochastic process (*Gillespie, 1977*). The IBM can be tailored to suit different cell types and experimental conditions by choosing different functions for $\psi_i$ and $\mu(\mathbf{x}_i, \mathbf{x}_i + \mathbf{r})$. In the following description, we choose functions suitable for simulating movement of fibroblast cells.

The movement rate $\psi_i$ comprises an intrinsic movement rate $m$ and a density-dependent component that sums contributions from $n$ neighbouring cells at $\mathbf{x}_j$ to individual $i$'s motility:

$$\psi_i = \max\left(0, m + \sum_{\substack{j=1 \\ i \neq j}}^{n} w(\mathbf{x}_j - \mathbf{x}_i)\right), \tag{1}$$

which ensures that $\psi_i \geq 0$. The kernel $w(\mathbf{z})$ weights the strength of interaction between a pair of cells displaced by $\mathbf{z}$ and for simplicity we choose it to be a Gaussian function

$$w(\mathbf{z}) = \alpha \exp\left(-\frac{|\mathbf{z}|^2}{2\sigma_w^2}\right). \tag{2}$$

The parameter $\alpha$ determines the interaction strength while $\sigma_w^2$ determines the range over which interactions occur.

We now describe a mechanism which allows a cell's direction of movement to be determined by the degree of crowding in its neighbourhood. This mechanism is comparable to that of *Binny, Plank & James (2015)* but with some differences that are required for extension to two spatial dimensions. The neighbour-dependent bias $\mathbf{b}(\mathbf{x})$ accounts for the effect of $n$ neighbouring cells located at $\mathbf{x}_j$ on the direction of movement of an individual at $\mathbf{x}$

$$\mathbf{b}(\mathbf{x}) = \sum_{j=1}^{n} \nabla v(\mathbf{x}_j - \mathbf{x}). \tag{3}$$
**A**

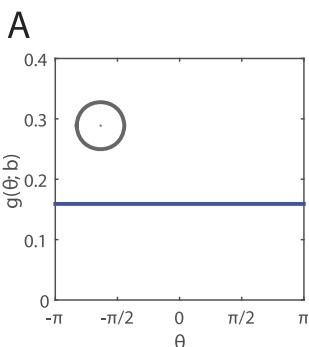

**B**

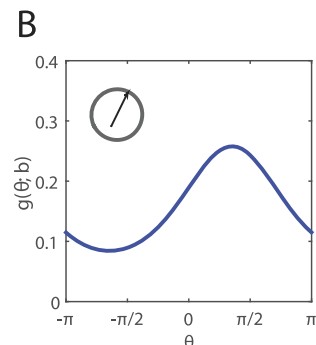

**C**

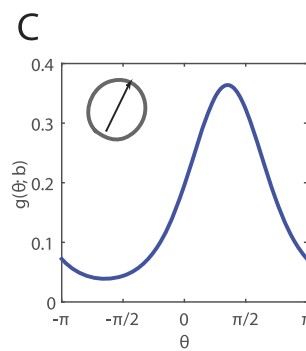

**Figure 1  Examples of probability density function $g(\theta; \mathbf{b})$ (blue solid line) for movement in a direction $\theta \in [0, 2\pi]$.** The neighbour-dependent bias $\mathbf{b}$ is a vector indicating the direction ($\arg(\mathbf{b})$) in which the greatest/lowest degree of crowding arises in a cell's neighbourhood, as well as the extent to which it occurs ($|\mathbf{b}|$). Insets are schematics illustrating $g(\theta; \mathbf{b})$ (grey solid line), where black arrows indicate the direction ($\arg(\mathbf{b})$) in which an individual (black dot) is most biased to move. (A) Unbiased movement; (B) weak directional bias $\mathbf{b} = (0.25, 0.5)^T$; (C) strong directional bias $\mathbf{b} = (0.5, 1)^T$.

The kernel $v(\mathbf{z})$ weights the strength of interaction between a cell pair displaced by $\mathbf{z}$. For simplicity, we choose $v(\mathbf{z})$ to be a Gaussian function

$$v(\mathbf{z}) = \beta \exp\left(-\frac{|\mathbf{z}|^2}{2\sigma_v^2}\right), \tag{4}$$

which means the interaction will be strong for a pair of cells located close together and negligible if they are far apart. Interaction strength and range are determined by $\beta$ and $\sigma_v^2$, respectively. The neighbour-dependent bias $\mathbf{b}(\mathbf{x})$ is a vector holding information about both the extent and direction of crowded regions in the neighbourhood of a cell at $\mathbf{x}$. We use the angle $\arg(\mathbf{b}(\mathbf{x}))$ to describe the direction of $\mathbf{b}(\mathbf{x})$. When $\beta > 0$, $\arg(\mathbf{b}(\mathbf{x}))$ is the direction in which the lowest degree of cell crowding arises locally. Conversely for $\beta < 0$, $\arg(\mathbf{b}(\mathbf{x}))$ is the direction of greatest local crowding. The magnitude $|\mathbf{b}(\mathbf{x})|$ provides a measure of the extent of crowding.

When a cell moves, its direction of movement $\theta \in [0, 2\pi]$ is drawn from a PDF $g(\theta; \mathbf{b})$ which depends on the neighbour-dependent bias $\mathbf{b}(\mathbf{x})$. The function $g(\theta; \mathbf{b})$ is a von Mises distribution with mean $\arg(\mathbf{b})$ and concentration $|\mathbf{b}|$:

$$g(\theta; \mathbf{b}) = \frac{\exp(|\mathbf{b}| \cos(\theta - \arg(\mathbf{b})))}{2\pi I_0(|\mathbf{b}|)}, \tag{5}$$

where $I_0$ is the modified Bessel function of order 0. Thus, a cell is most likely to move in the direction $\arg(\mathbf{b})$ and the strength of this directional bias increases with $|\mathbf{b}|$, as shown in Fig. 1.

The distance moved by a cell is drawn from a non-negative normal distribution with mean step length $1/\lambda_\mu$ and variance $\sigma_\mu^2$. Therefore, the probability of an individual at $\mathbf{x}$ moving to a new location at $\mathbf{y}$ is distributed according to

$$\mu(\mathbf{x}, \mathbf{y}) = N \exp\left(-\frac{\left(|\mathbf{y} - \mathbf{x}| - \frac{1}{\lambda_\mu}\right)^2}{2\sigma_\mu^2}\right) g(\arg(\mathbf{y} - \mathbf{x}); \mathbf{b}(\mathbf{x})). \tag{6}$$

This means that a cell at $\mathbf{x}$ is biased to move away from close-lying neighbours when $\beta > 0$. From a biological perspective this repulsive force could correspond to, for example, movement in response to a cell-released chemorepellant (*Cai, Landman & Hughes, 2006*) or physical forces due to deformation of the cell membrane under direct contact with other cells (*Trepat et al., 2009*). When $\beta < 0$ the bias is towards crowded regions, such as might arise in the presence of a cell-released chemoattractant (*Painter & Hillen, 2002*). The bias strength increases with increasing neighbourhood cell density. Setting $\beta = 0$ results in $g(\arg(\mathbf{y} - \mathbf{x}); \mathbf{b}(\mathbf{x})) = 1/(2\pi)$ and the cell is equally likely to move in any direction, i.e., movement is unbiased. The PDF $\mu(\mathbf{x}, \mathbf{y})$ has dimension $L^{-2}$ and normalising by the constant $N$ satisfies the constraint $\int \mu(\mathbf{x}, \mathbf{y}) d\mathbf{y} = 1$ for any fixed $\mathbf{x}$.

## Pair-correlation function

The second spatial moment, the average density of pairs of cells, can be expressed as a pair-correlation function (PCF) $C(r)$, written in terms of a separation distance $r$ (*Illian et al., 2008*). The PCF is normalised by dividing by the first moment squared such that $C(r) = 1$ in the complete absence of spatial structure, i.e., the distribution of cells is completely random (a Poisson spatial pattern). For $C(r) > 1$, pairs of cells are more likely to be found in close proximity than if they were distributed according to a Poisson pattern. We describe such a configuration of cells as a cluster spatial pattern. In contrast, for $C(r) < 1$, cell pairs separated by short displacements are less likely to arise, generating a regular spatial pattern.

We compute a PCF $C(r)$ from a particular arrangement of agents in a domain of width $L_x$ and height $L_y$. A reference agent at $\mathbf{x}_i$ is selected and the distance $r = |\mathbf{x}_j - \mathbf{x}_i|$ to a neighbour at $\mathbf{x}_j$ is calculated for $n - 1$ neighbours. A periodic PCF can be calculated by allowing a distance $r$ to be measured across periodic boundaries. A different reference agent is then chosen and the process repeated until each agent has been selected as a reference once. A PCF is constructed by counting the distances that fall into an interval $[r - \frac{\delta r}{2}, r + \frac{\delta r}{2}]$, i.e., binning distances using a bin width $\delta r$. To ensure $C(r) = 1$ in the complete absence of spatial structure we normalise by $n(n-1)(2\pi r \delta r)/(L_x L_y)$.

The choice of $\delta r$ is important because very small values can yield a PCF dominated by fluctuations while values that are too large result in an overly-smooth function which may mask spatial structure (*Binder & Simpson, 2015*).

## Spatial moment model

The IBM can be used to derive a population-level model in terms of the dynamics of spatial moments (*Plank & Law, 2015*). Mathematical descriptions of spatial moments and derivations of the rate of change equations for the first moment $Z_1(\mathbf{x}, t)$ and second moment $Z_2(\mathbf{x}, \mathbf{y}, t)$ are given in *Binny, Plank & James, (2015)* and still hold for movement in two dimensions. Spatial moments are functions of time as well as space but, for brevity, from here on we omit the time argument from the notation. Briefly, for the dynamics of the first spatial moment the corresponding description for $\psi_i$ is

$$M_1(\mathbf{x}) = m + \int w(\mathbf{y} - \mathbf{x}) \frac{Z_2(\mathbf{x}, \mathbf{y})}{Z_1(\mathbf{x})} d\mathbf{y}, \tag{7}$$

the expected movement rate of a cell at $\mathbf{x}$. In (1) a maximum formula ensured a non-negative movement rate but is not incorporated here because we only consider solutions in which negative expected movement rates do not arise. When a cell at $\mathbf{x}$ moves, its new location $\mathbf{y}$ is drawn from a PDF

$$\mu_1(\mathbf{x}, \mathbf{y}) = N \exp\left(-\frac{\left(|\mathbf{y} - \mathbf{x}| - \frac{1}{\lambda_\mu}\right)^2}{2\sigma_\mu^2}\right) g\left(\arg(\mathbf{y} - \mathbf{x}); \mathbf{b}_1(\mathbf{x})\right). \tag{8}$$

The neighbour-dependent bias for a cell at $\mathbf{x}$ is

$$\mathbf{b}_1(\mathbf{x}) = \int \nabla \nu(\mathbf{y} - \mathbf{x}) \frac{Z_2(\mathbf{x}, \mathbf{y})}{Z_1(\mathbf{x})} d\mathbf{y}. \tag{9}$$

The equation for the dynamics of the first spatial moment is

$$\frac{dZ_1(\mathbf{x})}{dt} = -M_1(\mathbf{x}) Z_1(\mathbf{x}) + \int \mu_1(\mathbf{u}, \mathbf{x}) M_1(\mathbf{u}) Z_1(\mathbf{u}) d\mathbf{u}, \tag{10}$$

where the first and second terms on the right-hand side correspond to movement out of $\mathbf{x}$ and into $\mathbf{x}$, respectively. The first moment is constant with respect to time because there are no birth/death events and there is no net flux across the boundaries.

For the dynamics of the second moment the expected movement rate of a cell at $\mathbf{x}$ in a pair with a cell at $\mathbf{y}$ is given by

$$M_2(\mathbf{x}, \mathbf{y}) = m + \int w(\mathbf{z} - \mathbf{x}) \frac{Z_3(\mathbf{x}, \mathbf{y}, \mathbf{z})}{Z_2(\mathbf{x}, \mathbf{y})} d\mathbf{z} + w(\mathbf{y} - \mathbf{x}), \tag{11}$$

where $Z_3(\mathbf{x}, \mathbf{y}, \mathbf{z})$ denotes the third spatial moment, the average density of triplets of cells. When a cell at $\mathbf{x}$ moves, its new location $\mathbf{y}$ is drawn from a PDF $\mu_2(\mathbf{x}, \mathbf{y}, \mathbf{z})$, where the third argument accounts for the fact that $\mathbf{x}$ is in a pair with a cell at $\mathbf{z}$:

$$\mu_2(\mathbf{x}, \mathbf{y}, \mathbf{z}) = N \exp\left(-\frac{\left(|\mathbf{y} - \mathbf{x}| - \frac{1}{\lambda_\mu}\right)^2}{2\sigma_\mu^2}\right) g\left(\arg(\mathbf{y} - \mathbf{x}); \mathbf{b}_2(\mathbf{x}, \mathbf{z})\right). \tag{12}$$

The neighbour-dependent bias for a cell at $\mathbf{x}$ in a pair with a cell at $\mathbf{y}$ is given by

$$\mathbf{b}_2(\mathbf{x}, \mathbf{y}) = \int \nabla \nu(\mathbf{z} - \mathbf{x}) \frac{Z_3(\mathbf{x}, \mathbf{y}, \mathbf{z})}{Z_2(\mathbf{x}, \mathbf{y})} d\mathbf{z} + \nabla \nu(\mathbf{y} - \mathbf{x}). \tag{13}$$

Finally, the equation for the dynamics of the second moment is

$$\begin{aligned}
\frac{dZ_2(\mathbf{x}, \mathbf{y})}{dt} &= -(M_2(\mathbf{x}, \mathbf{y}) + M_2(\mathbf{y}, \mathbf{x})) Z_2(\mathbf{x}, \mathbf{y}) \\
&\quad + \int \mu_2(\mathbf{u}, \mathbf{x}, \mathbf{y}) M_2(\mathbf{u}, \mathbf{y}) Z_2(\mathbf{u}, \mathbf{y}) d\mathbf{u} \\
&\quad + \int \mu_2(\mathbf{u}, \mathbf{y}, \mathbf{x}) M_2(\mathbf{u}, \mathbf{x}) Z_2(\mathbf{u}, \mathbf{x}) d\mathbf{u}. \tag{14}
\end{aligned}$$

Movement out of $\mathbf{x}$, conditional on the presence of a cell at $\mathbf{y}$, is accounted for in the first negative term in (14). The first integral term describes movement into $\mathbf{x}$ from a starting location $\mathbf{u}$, conditional on the presence of a cell at $\mathbf{y}$. The remainder are symmetric terms for movement out of and into $\mathbf{y}$.

A closure for the third spatial moment is required to solve Eq. (14) and we use the Kirkwood superposition approximation (*Kirkwood, 1935*; *Kirkwood & Boggs, 1942*) given by

$$\tilde{Z}_3(\mathbf{x},\mathbf{y},\mathbf{z}) = \frac{Z_2(\mathbf{x},\mathbf{y})Z_2(\mathbf{x},\mathbf{z})Z_2(\mathbf{y},\mathbf{z})}{Z_1(\mathbf{x})Z_1(\mathbf{y})Z_1(\mathbf{z})}, \tag{15}$$

however other choices of closure are possible (*Murrell, Dieckmann & Law, 2004*). This closes the dynamical system at second order, therefore we retain information on spatial structure that would be ignored by instead employing a first-order closure, such as the mean-field assumption.

## RESULTS

### Comparing IBM simulation data and moment dynamics approximations

To explore whether our model is capable of generating spatial structure in a simulated cell population we average results from repeated simulations of the IBM and compute a periodic PCF $C_{IBM}(r)$ as outlined in 'Pair-correlation function'. We compare this to numerical solutions of our spatial moment model to examine whether it provides a good approximation to the underlying stochastic process. The equation for the dynamics of the second moment (14) is solved for a spatially homogeneous distribution of cells, which means that we assume the probability of finding an individual in a given small region is independent of its location in space. This allows the equation to be rewritten in terms of displacements between pairs of cells, as outlined in the Appendix. The PCF $C_{SM}(\boldsymbol{\xi})$ is given by $Z_2(\boldsymbol{\xi})/Z_1^2$ such that $C_{SM}(\boldsymbol{\xi}) = 1$ in the complete absence of spatial structure. The second spatial moment is radially symmetric about the origin of $\boldsymbol{\xi}$. Therefore, in the results below we show only a radial section of $C_{SM}(\boldsymbol{\xi})$ which we denote $C_{SM}(r)$, where $r = |\boldsymbol{\xi}|$. Cells are initially distributed across a domain of width $L_x$ and height $L_y$, according to a spatial Poisson process with intensity $n/(L_x L_y)$. In the spatial moment model this corresponds to $Z_2(\boldsymbol{\xi}) = Z_1^2$ at $t = 0$. The system is allowed to reach steady state before results from each model are compared. Parameters used in this section are summarised in Table 1.

In the complete absence of interactions, an individual's direction of movement is unbiased and its movement rate is solely determined by the intrinsic component. It is straightforward to show analytically that the steady-state solution for $Z_2(\boldsymbol{\xi})$ is a constant under these conditions. Numerical solutions and averaged IBM simulations confirm this.

The effect of the neighbour-dependent directional bias, in the absence of neighbour-dependent motility (i.e., $\alpha = 0$), is shown in Fig. 2. The PCF quantifies differences in the spatial structure, depending on the strength and nature of cell–cell interactions, which may not be readily apparent from a qualitative visual inspection of the cell locations (Fig. 2 insets). Regular spatial patterns are generated by the directional bias when $\beta > 0$

**Table 1  Table of model parameters in order of appearance, with values used in the numerical results.**

| Symbol | Description | Units | Value Fig. 2 | Fig. 3 | Fig. 4 |
|---|---|---|---|---|---|
| $m$ | Intrinsic movement rate | h$^{-1}$ | 10 | 10 | 5 |
| $\alpha$ | Strength of interaction for movement rate | h$^{-1}$ | 0 | 1; 10; −1.5; −2 | 0 |
| $\sigma_w$ | Spatial range of interactions for movement rate | µm | 0.5 | 0.5 | 10 |
| $\beta$ | Strength of interaction for directional bias | µm | 0.1; 1; −0.03; −0.05 | 0 | 1,000 |
| $\sigma_v$ | Spatial range of interactions for directional bias | µm | 0.5 | 0.5 | 10 |
| $\lambda_\mu$ | Rate parameter of PDF for movement distance | µm$^{-1}$ | 5 | 5 | 0.1 |
| $\sigma_\mu$ | Spatial range of PDF for movement distance | µm | 0.05 | 0.05 | 2.5 |
| $\delta r$ | Bin width for PCF | µm | 0.12 | 0.12 | 8 |
| $\Delta$ | Grid spacing for discretisation of spatial displacement $\xi$ | µm | 0.1 | 0.1 | 5 |
| $\xi_{max}$ | Maximum distance of $\xi_1, \xi_2$ for computing $Z_2(\xi)$ | µm | 4 | 4 | 150 |

while $\beta < 0$ gives rise to clustering. The spatial moment model performs very well as an approximation to the IBM except when there is strong clustering (Fig. 2D). This can likely be attributed to limitations of the moment-closure assumption. The Kirkwood Superposition Approximation provides a reasonable approximation to the third moment for Poisson spatial patterns and regular patterns, but performs quite poorly for cluster spatial patterns where it can cause the model to underestimate the second moment (*Raghib, Hill & Dieckmann, 2011*; *Murrell, Dieckmann & Law, 2004*; *Dieckmann & Law, 2000*).

Figure 3 shows the spatial structure generated by the mechanism for neighbour-dependent motility when there is no local directional bias (i.e., $\beta = 0$). Neighbourhood interactions give rise to regular spatial patterns when $\alpha > 0$ and cluster spatial patterns when $\alpha < 0$. Again, we see good agreement between $C_{SM}(r)$ and $C_{IBM}(r)$ except for large magnitudes of $\alpha < 0$ where the pattern is clustered and the moment model under-predicts spatial structure (Fig. 3D). While the limitations associated with the moment closure may play a role, there is another factor that could also be contributing to the poor fit here. We have chosen values of $\alpha$ such that the probability of $\psi_i > 0$ is high. However $\psi_i = 0$ can arise by chance in an IBM simulation and while such occurrences are relatively rare they can have a self-propagating effect, leading to strong clustering. The spatial moment model does not account for these chance events so this might explain why spatial structure is underestimated more dramatically even for relatively weak clustering.

Our numerical results show that the same spatial structures can be generated by either neighbour-dependent mechanism acting in isolation. When both mechanisms affect movement together, the choice of $\alpha$ and $\beta$ determines whether they work cooperatively, to promote spatial structure to an even greater extent, or in opposition.

## Model validation using experimental data

We will now use *in vitro* experimental data to validate our model. We begin by exploring whether the directional bias mechanism is capable of generating spatial structure that is

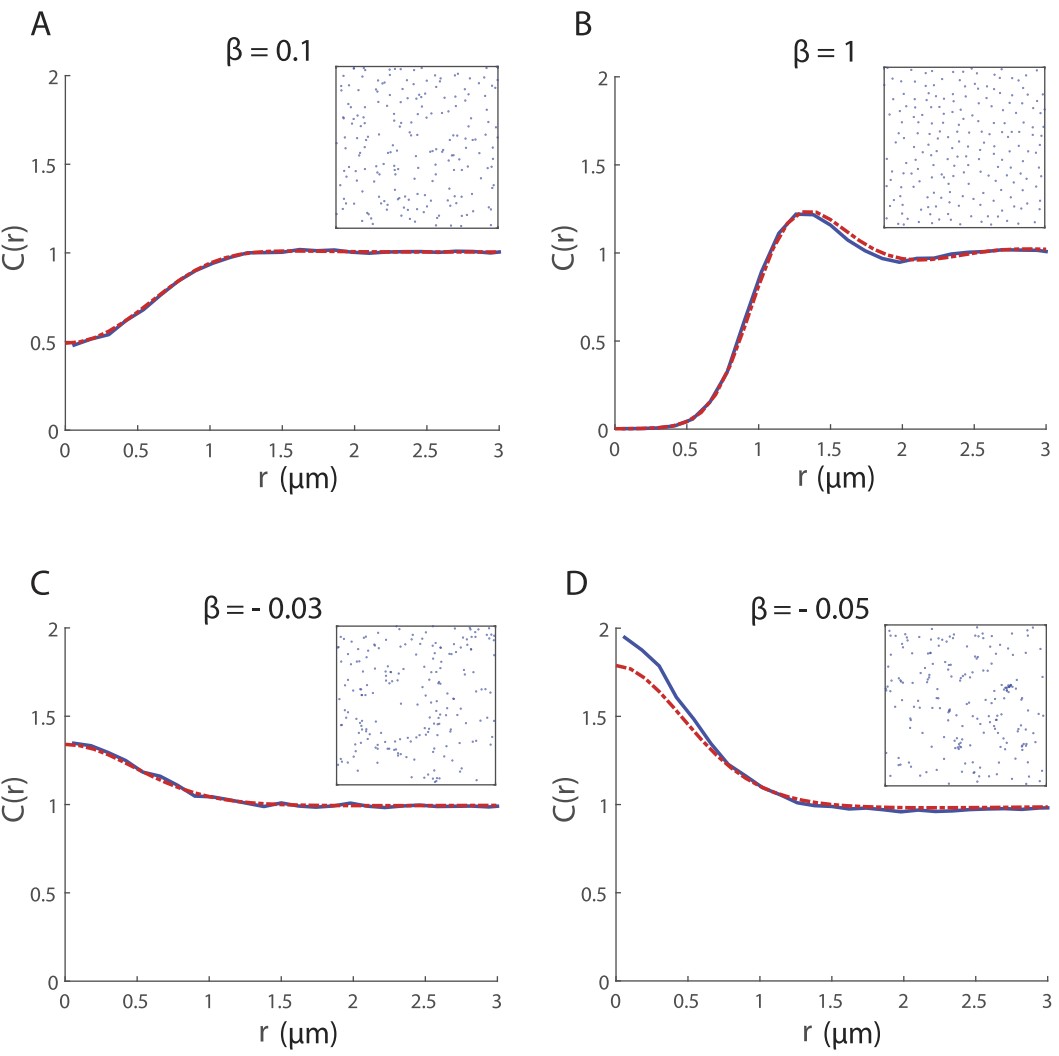

**Figure 2** **Spatial structure for 200 cells undergoing collective movement with neighbour-dependent directional bias ($\alpha = 0\,\mathrm{h}^{-1}$) in a 20 μm × 20 μm domain at time $t = 25\,\mathrm{h}$.** The PCF $C_{\mathrm{IBM}}(r)$ (blue solid line) provides a quantitative measure of the spatial structure in the simulated cell population and is computed (using a bin width $\delta r = 0.12\,\mu\mathrm{m}$) by averaging results from 500 repeated simulations of the IBM. For ease of visualisation, a snapshot of the configuration of cells in a single simulation at $t = 25$ is shown in the inset. The spatial structure approximated by the spatial moment model (solved using $\Delta = 0.1\,\mu\mathrm{m}$ and $\xi_{\max} = 4\,\mu\mathrm{m}$) is expressed as a PCF $C_{\mathrm{SM}}(r)$ (red dashed line). Parameters are $\alpha = 0\,\mathrm{h}^{-1}$, $\sigma_w = \sigma_v = 0.5\,\mu\mathrm{m}$, $m = 10\,\mathrm{h}^{-1}$, $\lambda_\mu = 5\,\mu\mathrm{m}^{-1}$, $\sigma_\mu = 0.05\,\mu\mathrm{m}$; (A) $\beta = 0.1\,\mu\mathrm{m}$; (B) $\beta = 1\,\mu\mathrm{m}$; (C) $\beta = -0.03\,\mu\mathrm{m}$; (D) $\beta = -0.05\,\mu\mathrm{m}$.

qualitatively similar to that observed in 3T3 fibroblast cell populations studied *in vitro* and aim to estimate parameters which yield a reasonable qualitative match to our data.

Movement rates for 3T3 fibroblast cells are discussed in the literature (*Ware, Wells & Lauffenburger, 1998*; *Vedel et al., 2013*). We choose a biologically relevant rate of 50 μm/h for the speed at which an isolated cell moves (i.e., in the absence of neighbourhood interactions). Cell speed is not itself a parameter of our model, but can be decomposed into two constituent parts for input into the model: a mean step length $1/\lambda_\mu = 10\,\mu\mathrm{m}$ and

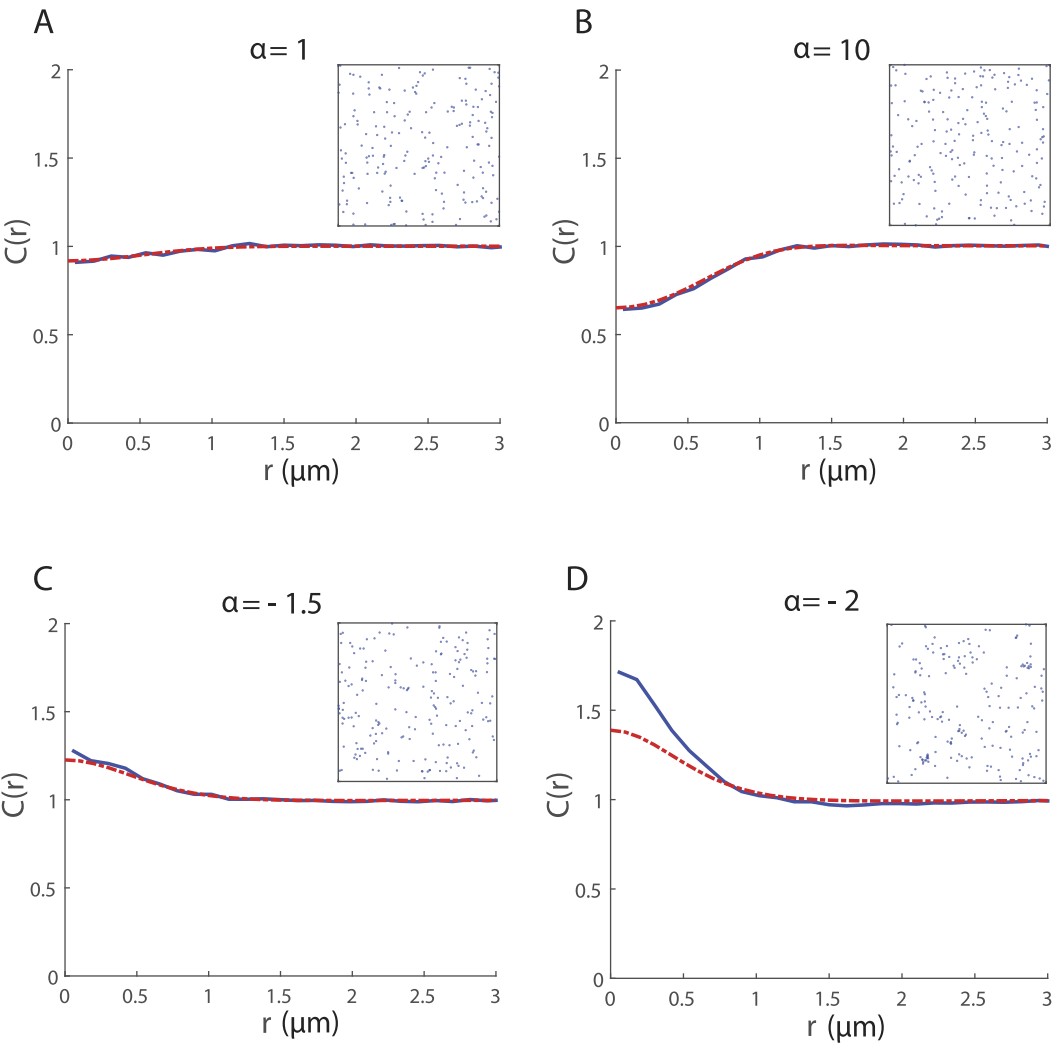

**Figure 3** **Spatial structure for 200 cells undergoing collective movement with neighbour-dependent motility ($\beta = 0$ µm) in a 20 µm × 20 µm domain at time $t = 25$ h.** The PCF $C_{IBM}(r)$ (blue solid line) provides a quantitative measure of the spatial structure in the simulated cell population and is computed (using a bin width $\delta r = 0.12$ µm) by averaging results from 500 repeated simulations of the IBM. For ease of visualisation, a snapshot of the configuration of cells in a single simulation at $t = 25$ is shown in the inset. The spatial structure approximated by the spatial moment model (solved using $\Delta = 0.1$ µm and $\xi_{max} = 4$ µm) is expressed as a PCF $C_{SM}(r)$ (red dashed line). Parameters are $\beta = 0$ µm, $\sigma_w = \sigma_v = 0.5$ µm, $m = 10$ h$^{-1}$, $\lambda_\mu = 5$ µm$^{-1}$, $\sigma_\mu = 0.05$ µm; (A) $\alpha = 1$ h$^{-1}$; (B) $\alpha = 10$ h$^{-1}$; (C) $\alpha = -1.5$ h$^{-1}$; (D) $\alpha = -2$ h$^{-1}$.

an intrinsic movement rate $m = 5$ h$^{-1}$. For the movement PDF $\mu(\mathbf{x}, \mathbf{y})$ we set $\sigma_\mu = 2.5$ µm which is biologically reasonable as it ensures cells are more likely to take short steps than undergo large jumps across the space. We employ the directional bias mechanism to incorporate volume exclusion effects by interpreting $2\sigma_v$ as the approximate range over which a cell interacts with neighbours and treating this as a proxy for the average diameter of a cell. From the literature, the average cell diameter for 3T3 fibroblast cells is approximately 20 µm which yields $\sigma_v = 10$ µm (*Simpson et al., 2013a*; *Vedel et al., 2013*). Here, we consider the directional bias mechanism in the absence of neighbour-dependent

motility (i.e., we set $\alpha = 0$). With these parameter choices in place, interaction strength $\beta$ is the only parameter that we need to estimate.

Images are taken at the centre of the well to avoid edge effects and when analysing our *in vitro* data, we assume that cells are distributed homogeneously across this region. An average cell density is estimated from each image, by dividing the number of cells in an image (which ranged between 80 and 318 cells) by the image area. In 'Comparing IBM simulation data and moment dynamics approximations' we implemented periodic boundary conditions in our IBM simulations such that cells located near a boundary of the domain could interact with those at an opposite boundary. Therefore it was reasonable to calculate a periodic PCF from the configurations of cells that arose. However, for our experimental data, the motility of a cell located near the edge of an image will not be affected by a cell at an opposite edge. Therefore, to calculate an accurate average pair density for the short displacements we are primarily interested in, we choose to generate a non-periodic PCF $C_{exp}(r)$ from the experimental images.

To obtain an estimate for $\beta$ we consider a single experimental image of dimensions 4,500 µm $\times$ 450 µm with 286 cells, as shown in Fig. 4A with markers superimposed over cell locations. We use our IBM to simulate movement in this 4,500 µm $\times$ 450 µm region using the parameters discussed above (and summarised in Table 1) and explore different values of $\beta$. In each simulation, 286 cells are initially distributed according to a spatial Poisson process and we compute a PCF once the system has converged to steady state. Figure 4B shows a snapshot from an IBM simulation at $t = 15$ h. The presence of spatial structure is not obvious from visual inspection of Figs. 4A–4B alone but calculating a PCF (Fig. 4C) indicates a regular spatial pattern over displacements <50 µm. We find that for $\beta = 1,000$ µm the PCFs predicted by our IBM and spatial moment model provide a very good visual match to that computed from the *in vitro* data for this sample. Unlike $C_{IBM}(r)$ and $C_{SM}(r)$, the PCF computed from each experimental image does not tend to 1 for large displacements because it is computed from non-periodic distances and owing to the image dimensions. However, we see good agreement at short to moderate displacements. To validate our estimate, we compare PCFs obtained using the same parameter choices and $\beta = 1,000$ µm for the average cell densities in each of the other images (Figs. S1 and S2). For all samples we see a reasonable qualitative agreement between the PCFs predicted by the model and the PCF generated from the *in vitro* data.

The PCFs $C_{exp}(r)$ and $C_{IBM}(r)$ employ a bin width $\delta r$ which provides a reasonably smooth function for the majority of experimental samples yet contains sufficient information about spatial structure to allow us to carry out our analysis. Smaller values of $\delta r$ give a better match to $C_{SM}(r)$, however $C_{exp}(r)$ becomes dominated by fluctuations.

From our numerical results we know that both the mechanisms for neighbour-dependent motility and directional bias are capable of generating spatial structure. In the absence of directional bias, large values of $\alpha$ are required to generate the extent of spatial structure observed in the *in vitro* data. When carrying out IBM simulations under these conditions, individuals experience strong neighbourhood interactions and, as a result, movement rates $\psi_i$ are often considerably higher than the average movement rates of fibroblast cells discussed in the literature (*Ware, Wells & Lauffenburger, 1998*; *Vedel et al., 2013*). For

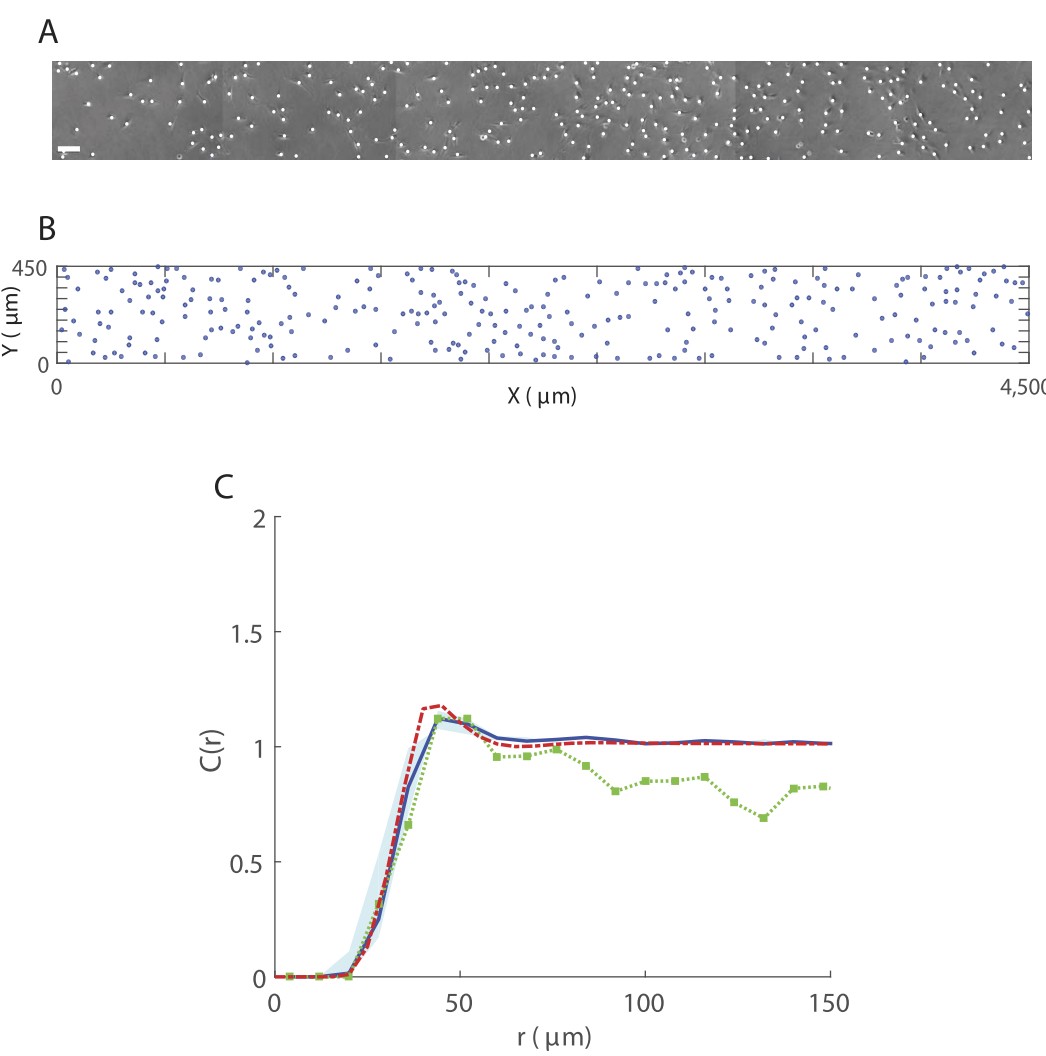

**Figure 4  Spatial structure in 3T3 fibroblast cells for 286 cells in a 4,500 μm × 450 μm region.** (A) Sample image (obtained from a well containing cell suspension of approximate initial density 30,000 cells/ml) showing superimposed markers (white dots). Scale bar corresponds to 100 μm; (B) Cell locations (blue dots) at $t = 15$ h from a single IBM simulation. Parameters are $\alpha = 0$ h$^{-1}$, $\beta = 1,000$ μm, $\sigma_w = \sigma_v = 10$ μm, $m = 5$ h$^{-1}$, $\lambda_\mu = 0.1$ μm$^{-1}$, $\sigma_\mu = 2.5$ μm; (C) PCF $C_{\text{IBM}}(r)$ (blue solid line) obtained from averaging results from 200 simulations of the IBM at $t = 15$ h. PCFs computed from the IBM using values of $\beta$ within the range $\pm 75\%$ of $\beta = 1,000$ μm, lie within the region indicated by the blue shaded area. PCF $C_{\text{exp}}(r)$ (green squares-dotted line) generated from experimental image, for $\delta r = 8$ μm. PCF $C_{\text{SM}}(r)$ (red dashed line) approximated by spatial moment model at $t = 15$ h, for $\Delta = 5$ μm and $\xi_{\text{max}} = 150$ μm.

example, using the same parameter choices as for Fig. 4 but in the absence of directional bias ($\beta = 0$), an interaction strength of $\alpha = 1,000$ h$^{-1}$ generates spatial structure which is a reasonable qualitative match to the *in vitro* data. However, 23% of individuals undergo movement with a rate $\psi_i > 100$ h$^{-1}$, which corresponds to a biologically unreasonable cell speed of 1,000 μm/h. Therefore, we do not consider neighbour-dependent motility in isolation here. When both mechanisms are acting together, numerous combinations of $\alpha$ and $\beta$ exist that would give rise to similar spatial structure.

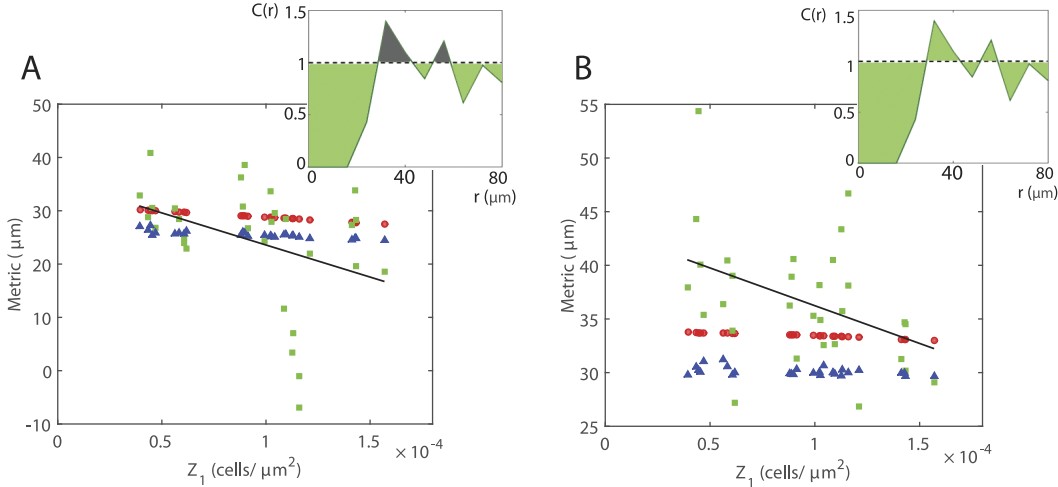

**Figure 5** **Relationship between average cell density and the extent of spatial structure.** Metrics calculated from IBM (blue triangles), spatial moment model (red circles) and *in vitro* data (green squares) for the average cell densities in each of the images. A regression line (black line) is fitted to the experimental data. (A) Metric calculated by integrating $(1 - C(r))$ over displacements $0 \leq r \leq 80$ μm, i.e., summing the green-shaded area and subtracting the grey-shaded area (inset Fig.). (B) Metric calculated by integrating $|1 - C(r)|$ over displacements $0 \leq r \leq 80$ μm, i.e., summing the green-shaded area (inset Fig.).

Numerical and analytical results suggest that there is a relationship between the average cell density and the extent of spatial structure in the moving cell population. Increasing the average cell density causes a decrease in the extent of spatial structure, i.e., for a regular spatial pattern average pair densities at short displacements increase towards 1. However, for the average cell densities studied here, it is not immediately obvious whether our *in vitro* experimental data supports the suggestion that a significant relationship exists. We now explore this idea in more depth by using the area between the PCF to calculate a summary statistic which quantifies the extent of spatial structure, as shown in Fig. 5. We consider two metrics and compute each for PCFs generated from the IBM, spatial moment model and *in vitro* data. The first metric measures spatial structure as $\int_0^R (1 - C(r)) \mathrm{d}r$ (Fig. 5A). Positive values indicate a regular spatial pattern while negative values indicate a cluster spatial pattern. The second is given by $\int_0^R |1 - C(r)| \mathrm{d}r$ (Fig. 5B). Both metrics are calculated for $R = 80$ μm and have units μm. The average cell densities obtained from the *in vitro* data lie within a relatively small range and so the overall change in the metric is small. Nevertheless, for both metrics our model predicts that increasing average cell density decreases the extent of spatial structure. To investigate whether our *in vitro* data supports this we carry out a simple linear regression, yielding *p*-values of 0.0211 and 0.0435 for the first (Fig. 5A) and second metric (Fig. 5B), respectively. Thus, using either metric and despite the noise in our *in vitro* data, the results suggest that a significant relationship does indeed exist between average cell density and spatial structure.

## DISCUSSION

IBMs of collective movement allow us to explore how interactions between individuals give rise to spatial structure and how, in turn, this self-generated spatial structure affects the population dynamics. However, IBMs are limited when it comes to explaining population-level behaviour as they can be difficult to analyse mathematically. To move beyond these limitations, population-level models can be derived from IBMs but often employ a mean-field assumption which neglects spatial correlations between cells. We have derived a population-level description in terms of spatial moment dynamics to account for spatial correlations and give insight into how neighbour-dependent directional bias generates spatial structure in a moving cell population. Extending our original model (*Binny, Plank & James, 2015*) from one to two spatial dimensions makes it more amenable for use alongside experimental data. Our results verify that the spatial moment model can provide a good approximation to averaged simulations of the underlying IBM when cells are distributed homogeneously through space.

Volume exclusion effects can be incorporated into lattice-free models of interacting agents, for example using a hard sphere approach where neighbours are explicitly excluded from a region surrounding an individual. Instead, we employ the mechanism for neighbour-dependent directional bias as a means of accounting for crowding effects. Using an interaction kernel concentrated around short pair displacements allows us to reduce the likelihood of two cells being found in very close proximity, although it does not altogether rule out the possibility.

*In vitro* studies have shown that cell motility can be heavily influenced by the average density of cells, particularly at high densities where crowding effects come into play, affecting the movement rate or direction of individuals (*Lee, McIntire & Zygourakis, 1994*; *Tremel et al., 2009*; *Vedel et al., 2013*). In addition, spatial correlations between cells can have major implications for motility, for example cell populations with clustering exhibit different behaviour to those that adopt regular spatial patterns (*Green et al., 2010*; *Keeley et al., 2014*). We carried out *in vitro* experiments with motile 3T3 fibroblast cells for model validation and to explore the extent to which spatial structure is generated in fibroblast cell populations. It is not obvious from visual inspection of the imaging data alone whether spatial structure is present, however calculating a PCF indicates a regular spatial pattern. The spatial structure arises over displacements <50 µm and is likely predominantly a consequence of space being excluded by the cells, however chemotactic interactions, such as chemokine signalling, may also contribute to a lesser extent (*Vedel et al., 2013*). We consider whether our model's mechanism for neighbour-dependent directional bias can generate a similar spatial structure. The majority of model parameters are obtained by selecting biologically relevant values from the literature and we use our *in vitro* data to provide an estimate for the interaction strength $\beta$. This parameter was estimated from a single experimental image and for validation we use the same estimate for the average cell densities in each of the other images. A visual comparison of the PCFs suggests that our parameterised model can successfully predict the spatial structure of 3T3 fibroblasts at various average cell densities. We do not consider the neighbour-dependent motility

mechanism in the absence of directional bias because the spatial structure observed *in vitro* could only be generated if a large proportion of cells moved at biologically unreasonable rates. However, it is possible that both mechanisms acting together could give rise to the observed spatial structure and further information would be required to distinguish the relative contributions of each effect occurring *in vitro*.

We choose to calculate a non-periodic PCF from each experimental image to obtain an accurate average pair density at short displacements. Because we do not apply edge corrections and owing to the image dimensions, the PCF often has values less than 1 for large displacements. However, we would expect that a PCF calculated either for a very large number of cells (at the same average density) or by averaging results from many identically-prepared repeated experiments, would give $C(r) \approx 1$ for large displacements. A number of methods to account for edge effects are discussed in the literature, for example the use of buffer zones, toroidal edge corrections or employing weighting factors (*Haase, 1995*; *Law et al., 2009*). However, in some cases, applying an edge correction may yield results that do not provide an accurate representation of the spatial structure in the population. For instance, when analysing spatial patterns that are clustered or regular, the use of a toroidal correction can lead to an unknown extent of bias in the resulting distribution of distances (*Haase, 1995*). To avoid this uncertainty, we have chosen to work with the actual pair distances between cells in the experimental images and not correct for edge effects.

We have further validated our model by considering in more detail the relationship between average cell density and the extent of spatial structure in a cell population. Numerical and analytical results from our model suggest that increasing the average cell density decreases the extent of spatial structure. There is considerable noise in the *in vitro* data because we choose to analyse PCFs generated from individual images as opposed to working with averaged results. In addition, the data considers a relatively small range of average cell densities. Nevertheless, our experimental data also supports the idea that such a relationship exists. The most likely explanation for this effect is that as average cell density increases, there is less free space available and cells are forced into closer proximity. Because of their deformable plasma membranes, pairs of cells can arise at displacements less than the average diameter of a cell. This increases the average pair density at short displacements, thus reducing the extent of spatial structure. Because we do not employ a hard sphere volume-exclusion method, instead representing cells by points in space, our model will predict a Poisson spatial pattern for very high average cell densities (far greater than those in our data). In reality, the fact that 3T3 fibroblasts have a minimum area they can occupy means that this would never be observed *in vitro*.

The spatial moment model is only an approximation to the IBM because it invokes a closure assumption which closes the dynamical system at second order and ignores higher order moments. The performance of our model depends on the suitability of this closure as an approximation to the third moment. Different closures are proposed in the literature and we use the Kirkwood Superposition Approximation, which is a relatively simple closure that is often applied in cell movement models. This closure is known to perform reasonably well for regular and Poisson spatial patterns but causes the model to underestimate the
second moment for cluster patterns. A number of other closures also share this limitation. The asymmetric power-2 closure, which expresses the third moment in terms of weighted sums of lower order moments, can prove more successful for cluster spatial patterns. However it is not always obvious which weighting constants are most appropriate and the closure has the potential to predict negative average densities of triplets (*Dieckmann & Law, 2000*; *Murrell, Dieckmann & Law, 2004*; *Raghib, Hill & Dieckmann, 2011*).

We have chosen to use kernels suitable for modelling fibroblast movement but different kernels could be employed for applications in other contexts. However, there is a numerical constraint associated with choosing the movement PDF $\mu$. If using a PDF that has large positive values concentrated at pair displacements very close to zero, the spatial moment model cannot always accurately capture the full extent of the directional bias at these short displacements. This, in turn, causes the model to underestimate the extent of spatial structure. Choosing a movement PDF with positive values at displacements further from zero, such as the PDF employed here, overcomes this issue. Expressing and solving the moment dynamics equations in polar coordinates may also allow for greater flexibility in the choice of movement PDF.

There are a number of possible extensions to the work presented here. For example, the model could be extended to a birth-death-movement process to investigate how cell proliferation and cell death contribute to the collective dynamics. Models of spatial moment dynamics that incorporate density-independent or density-dependent birth, death and movement have previously been discussed in the literature (see for example *Dieckmann & Law, (2000)*; *Murrell (2005)*) but it would be useful to explore the role that neighbour-dependent directional bias plays in this setting. We have applied our model to cell movement, however the types of interaction experienced by cells are also relevant in other contexts. For instance, our model could be applied in an ecological context to consider the effect of directional bias on moving animal populations.

## APPENDIX

When solving Eq. (14) for a spatially homogeneous distribution of cells, the second moment $Z_2(\mathbf{x}, \mathbf{y})$ depends only the displacement $\mathbf{y} - \mathbf{x}$ which can now be treated as a single variable. The displacement from $\mathbf{x}$ to $\mathbf{y}$ is denoted $\boldsymbol{\xi}$ and the displacement from $\mathbf{x}$ to $\mathbf{z}$ is denoted $\boldsymbol{\xi}'$. For the movement PDF $\mu_2(\mathbf{u}, \mathbf{x}, \mathbf{y})$, we denote the displacement from $\mathbf{u}$ to $\mathbf{x}$ as $\boldsymbol{\xi}''$. The first spatial moment is required for $\tilde{Z}_3(\mathbf{x}, \mathbf{y}, \mathbf{z})$ and in the homogeneous case $Z_1$ is a constant.

We rewrite (14) in terms of the displacements between pairs as follows:

$$
\begin{aligned}
\frac{\mathrm{d}Z_2(\boldsymbol{\xi})}{\mathrm{d}t} = & -(M_2(\boldsymbol{\xi}) + M_2(-\boldsymbol{\xi}))Z_2(\boldsymbol{\xi}) \\
& + \int \mu_2(\boldsymbol{\xi}'', \boldsymbol{\xi}'' + \boldsymbol{\xi})M_2(\boldsymbol{\xi}'' + \boldsymbol{\xi})Z_2(\boldsymbol{\xi}'' + \boldsymbol{\xi})\mathrm{d}\boldsymbol{\xi}'' \\
& + \int \mu_2(\boldsymbol{\xi}'', \boldsymbol{\xi}'' - \boldsymbol{\xi})M_2(\boldsymbol{\xi}'' - \boldsymbol{\xi})Z_2(\boldsymbol{\xi}'' - \boldsymbol{\xi})\mathrm{d}\boldsymbol{\xi}''.
\end{aligned}
\tag{16}
$$

The movement rate $M_2(\mathbf{x}, \mathbf{y})$ of a cell at $\mathbf{x}$ in a pair with a cell at $\mathbf{y}$ given in (11) is now expressed in terms of the displacement $\boldsymbol{\xi}$ between $\mathbf{x}$ and $\mathbf{y}$:

$$M_2(\boldsymbol{\xi}) = m + \int w(\boldsymbol{\xi}') \frac{Z_3(\boldsymbol{\xi}, \boldsymbol{\xi}')}{Z_2(\boldsymbol{\xi})} d\boldsymbol{\xi}' + w(\boldsymbol{\xi}). \tag{17}$$

The movement PDF given in (12) becomes

$$\mu_2(\boldsymbol{\xi}, \boldsymbol{\xi}') = N \exp\left(-\frac{\left(|\boldsymbol{\xi}| - \frac{1}{\lambda_\mu}\right)^2}{2\sigma_\mu^2}\right) g(\arg(\boldsymbol{\xi}); \mathbf{b_2}(\boldsymbol{\xi}')) \tag{18}$$

with neighbour-dependent bias

$$\mathbf{b_2}(\boldsymbol{\xi}) = \int \nabla v(\boldsymbol{\xi}') \frac{Z_3(\boldsymbol{\xi}, \boldsymbol{\xi}')}{Z_2(\boldsymbol{\xi})} d\boldsymbol{\xi}' + \nabla v(\boldsymbol{\xi}). \tag{19}$$

The interaction kernels were previously expressed in terms of a single variable in (2) and (4) and these definitions still hold here. The closure for the third moment is

$$\tilde{Z}_3(\boldsymbol{\xi}, \boldsymbol{\xi}') = \frac{Z_2(\boldsymbol{\xi}) Z_2(\boldsymbol{\xi}') Z_2(\boldsymbol{\xi}' - \boldsymbol{\xi})}{Z_1^3}. \tag{20}$$

The boundary condition is as follows:

$$Z_2(\boldsymbol{\xi}) \to Z_1^2 \quad \text{as } |\boldsymbol{\xi}| \to \infty. \tag{21}$$

Equation (16) was solved numerically using the method of lines with MATLAB's in-built ode23 solver. This involved a discretisation of $\boldsymbol{\xi} = (\xi_1, \xi_2)^T$ with grid spacing $\Delta$ over the domain $\{-\xi_{max} \leq \xi_1, \xi_2 \leq \xi_{max}\}$, where $\xi_{max}$ was large enough so that $Z_2(\boldsymbol{\xi}) \approx Z_1^2$ at the boundary. Required values of $Z_2(\boldsymbol{\xi})$ that lay outside of the computable domain were set to the value of $Z_2(\boldsymbol{\xi})$ at a corner of the boundary, i.e., $Z_2(\xi_{max}, \xi_{max})$. The integral terms in (16) were approximated using the trapezium rule with the same discretisation. In addition, the PDF for movement $\mu_2(\boldsymbol{\xi}, \boldsymbol{\xi}')$ was normalised numerically using the trapezium rule such that $\int \mu_2(\boldsymbol{\xi}, \boldsymbol{\xi}') d\boldsymbol{\xi} = 1$ for any fixed $\boldsymbol{\xi}'$. The results were insensitive to a reduction in grid spacing $\Delta$.

### Funding

This research was supported by the Royal Society of New Zealand Marsden Fund, grant no. 11-UOC-005. The funders had no role in study design, data collection and analysis, decision to publish, or preparation of the manuscript.

### Grant Disclosures

The following grant information was disclosed by the authors:
Royal Society of New Zealand Marsden Fund: 11-UOC-005.

## Competing Interests

The authors declare there are no competing interests.

## Author Contributions

- Rachelle N. Binny conceived and designed the experiments, performed the experiments, analyzed the data, wrote the paper, prepared figures and/or tables, reviewed drafts of the paper.
- Parvathi Haridas conceived and designed the experiments, performed the experiments, reviewed drafts of the paper.
- Alex James and Richard Law analyzed the data, reviewed drafts of the paper.
- Matthew J. Simpson conceived and designed the experiments, contributed reagents/materials/analysis tools, reviewed drafts of the paper.
- Michael J. Plank conceived and designed the experiments, analyzed the data, reviewed drafts of the paper.

## Data Availability

Raw experimental data and code for mathematical model are provided in the Supplemental Information 1.

## Supplemental Information

Supplemental information for this article can be found online at http://dx.doi.org/10.7717/peerj.1689#supplemental-information.

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
