# Peer review of "Spatial structure arising from neighbour-dependent bias in collective cell movement"

_PeerJ, doi:10.7717/peerj.1689_

## Round 0.1 · original submission · Major Revisions

Please carefully take into account the reviewers comments in your revision.

Reviewer 1 ·

Basic reporting

According to PeerJ policies, the submitted manuscript should be accompanied by a code used to implement the mathematical model - this requirement has not been fulfilled.

Experimental design

No comments

Validity of the findings

When comparing all IBM images in the figure insets (except of panels D in Figs. 2 and 3), all simulated cell distributions look qualitatively similar, thus this is not clear whether the proposed C_IBM and C_SM can distinguish between these cell configurations. Some quantitative comparison between the obtained numerical results should be included to validate the techniques that have been presented.

Additional comments

The supplemental material (over 50 individual images) is impossible to review -- this should be collected in one file with proper annotations or legends.

·

Basic reporting

Pass.

This is a clearly and thoughtfully written piece of scholarship that describes the authors step toward a comprehensive method of understanding collective cell movement. In this manuscript the authors utilize an individual based model and analytic methods to predict the eventual structure (after movement with graded levels of interaction with neighbors) of cellular populations.

They show that without the effect of nearby cells, the structure that they observe in their experimental construct does not match the theoretical predictions. This work stands alone as a single piece of scholarship, but also moves forward nicely from their earlier work in 1-dimensional movement.

Experimental design

Pass.

The experimental design was clearly described, as were the mathematical methods. Where there were gaps in the explanation, there were clear references to earlier work where these could be filled in. There were no ethical issues with this research.

Validity of the findings

I have no doubt that the findings in this work were reported dutifully, and that the methods were carried out as described.

There was discussion of neglecting the edge effects, and in particular only using a (relatively) small field of view - further, in the analytic methods, periodic boundary conditions were employed. I do wonder how the application of an edge correction method would (if at all) change the analysis.

---

## Round 0.2 · accepted · Accept

Thank you for revising the paper in accordance with the reviewers' suggestions.

Reviewer 1 ·

Basic reporting

the authors have responded to my comments in a satisfactory way

Experimental design

no comments

Validity of the findings

the authors have responded to my comments in a satisfactory way

Additional comments

the authors have responded to my comments in a satisfactory way

·

Basic reporting

Pass.

Experimental design

No Comments.

Validity of the findings

No Comments.

Additional comments

I am happy with the revised manuscript and appreciate the authors efforts to improve what was already a solid piece of work.